# Genome-Wide Analysis of UGT Genes in Petunia and Identification of *PhUGT51* Involved in the Regulation of Salt Resistance

**DOI:** 10.3390/plants11182434

**Published:** 2022-09-19

**Authors:** Lili Dong, Ziyan Tang, Tianyin Yang, Fuling Hao, Xinyi Deng

**Affiliations:** College of Horticulture, Anhui Agricultural University, Hefei 230036, China

**Keywords:** petunia, salt resistance, UGT, expression analysis, *PhUGT51*, functional analysis

## Abstract

UDP-glycosyltransferase (UGT) plays an essential role in regulating the synthesis of hormones and secondary metabolites in plants. In this study, 129 members of the Petunia UGT family were identified and classified into 16 groups (A–P) based on phylogenetic analysis. The same subgroups have conserved motif compositions and intron/exon arrangement. In the promoters of the Petunia UGT genes, several cis-elements associated with plant hormones, growth and development, and abiotic stress have been discovered. Their expression profiles in five tissues were revealed by tissue expression based on RNA-seq data. Subcellular localization analysis showed that PhUGT51 was located in the nucleus and cell membrane. Salt stress caused an increase in the expression level of *PhUGT51*, but the expression level remained stable with the growth over time. In addition, the overexpression of *PhUGT51* caused a significant increase in salt resistance. Our study systematically analyses the UGT gene family in Petunia for the first time and provides some valuable clues for the further functional studies of UGT genes.

## 1. Introduction

Soil salinization is one of the most important abiotic stresses affecting plant growth, which seriously affects the yield of crops and the planting range of ornamental plants. Salt stress will produce ionic toxicity to plants, affect the absorption of water by plants, destroy physiological mechanisms, and eventually lead to plant wilting and death [1]. To survive under salt stress, plants have evolved various complex mechanisms to reduce the damage caused by salt stress [2]. Therefore, understanding the molecular mechanism of salt stress is of great significance in improving crop salt tolerance.

UGT is a family of enzymes found in various organisms, including plants, mammals, fungi, bacteria, and viruses [3]. The sugar acceptors’ stability, transport, storage, reactivity, and bioactivity can all be impacted by glycosylation [4]. The glycosylated products are crucial for maintaining plant hormone balance [5]. Of the 106 carbohydrate glycosyltransferase (GT) families currently documented in the carbohydrate-active enzyme database, plant UGTs are categorised into family 1 [6]. A conserved motif known as Plant Secondary Product Glycosyltransferase may be found in the C-terminal area of UGTs, and its sequences at the N-terminal region are thought to be responsible for identifying a range of substrates (PSPG). A unique, highly conserved region of 44 amino acids known as the PSPG box has been discovered in all the investigated plants’ UGTs [7]. 

The UGT genes in the plant world were first found in Maize [8]. Subsequently, the functions of a lot of UGT genes in plants were identified [9,10,11,12,13]. These studies showed that UGT genes widely participated in regulating plant growth development and response to abiotic and biotic stress. UGT74E2, a member of the L subclass of the UGT family [14], is an auxin glucosyltransferase mainly using IBA as substrate. A previous study showed that the overexpression of *UGT74E2* increased the tolerance to drought and salt stress, indicating that *UGT74E2* regulates shoot branching and stress responses [15]. However, the function of *UGT74E1*, which is in the same subfamily with *UGT74E2*, is not clear. In addition, with the completion of the genome sequencing of multiple species, a lot of UGT family genes were identified, such as 180, 181, 182, 145, and 168 UGTs members were identified from rice [16], grape [17], soybean [18], pomelo [19], and peach [20], respectively. Whereas UGT members in Petunia have not been investigated.

*Petunia hybrida* Vilm is widely used in gardens and is an essential ornamental plant [21]. Soil salinization seriously affects the planting of Petunia; therefore, cultivating new varieties with a strong salt resistance is necessary for Petunia breeding. This study identified UGT proteins from the Petunia genome database and named them in an orderly way. Moreover, a comprehensive analysis of PhUGT genes was accomplished, including their phylogenetic relationship, exon–intron structure, conserved domains, cis-regulatory analysis and expression patterns in various tissues, subcellular localization, and the function of *PhUGT51* were analyzed. Through these analyses, we hoped to identify all the UGT genes in Petunia and to understand the process of PhUGTs in salt resistance.

## 2. Results

### 2.1. Identification and Physicochemical Properties Analysis of PhUGT Family Members 

The conserved PSPG box sequence of the UGT family was used for a BLASTp search in the Petunia genome database. Two hundred and forty-eight sequences were obtained preliminarily, and 129 sequences were obtained after removing 119 sequences that were redundant or lacked the PSPG domain. After that, the Petunia UGT family members were named (PhUGT1–PhUGT129). The predicted molecular weight of 129 PhUGTs ranged from 33.89 kDa (PhUGT106) to 64.47 kDa (PhUGT79), averaging 51.35 kDa. The length of UGT protein ranged from 298 aa (PhUGT106) to 573 aa (PhUGT79), with an average of 456 aa. PI values ranged from 4.69 (PhUGT7) to 9.37 (PhUGT15), averaging 5.92. Subcellular localization prediction results showed that most of the PhUGT genes were located on the microsomes (Appendix A).

### 2.2. Phylogenetic Analysis of UGT Family in Petunia 

The phylogenetic tree was constructed from 129 members of the PhUGT family and 21 members of the AtUGT family (Figure 1). A total of 16 groups of the UGT family were detected, which were named A–P, based on identified group in Arabidopsis. There were various numbers of UGTs in each group. Twenty-seven UGT members comprised group A, the largest group, while groups B, J, and N, had just one member. Groups C, D, E, F, G, H, I, K, L, and M had 2, 12, 20, 2, 8, 3, 5, 6, 10, and 5 members, respectively. Our analysis discovered two new groups, O and P, with 22 and 4 PhUGT members, respectively. 

### 2.3. Analysis of the Petunia UGT Protein Motifs and Gene Structure

In order to systematically investigate the distribution of conserved motifs and exon–intron in Petunia UGT members, meme software was used to analyze the conserved domain of PhUGT proteins. A total of 10 motifs appeared and motif 1 was the PSPG box domain. Motif 3, 9, 6, 2, 5, 4, 8, and 10 existed in most of the PhUGT proteins, but some motifs only existed in specific groups, such as motif 5, which mainly existed in group A and I. Moreover, motif 3 was in the beginning and motifs 8 and 10 were in the tail of most of the PhUGT sequences. 

We mapped the genomic makeup of each PhUGT gene to gain a deeper understanding of the evolution of the UGT gene family in Petunia. The PhUGT genes had intron counts ranging from 0 to 7. One to three introns were present in most of the PhUGT genes, with fifty-three genes having just one intron, ten genes having two introns, and four genes having three introns. Additionally, there were no introns in the remaining 57 PhUGT genes. As seen in Figure 2, most PhUGT genes belonging to the same group had a comparable gene structure, for example, all group K members contained one intron, demonstrating the functional affinity shared by various members.

### 2.4. Analysis of Cis-Elements in the Promoters of PhUGT Genes

To comprehend the peculiarities of the distribution of cis-acting elements in the promoter sequence of PhUGT, 16 different types of cis-acting elements in UGT promoters were computed and compared (Figure 3). The PhUGT gene promoters were most frequently found to have the light responsive elements (Box 4, AE-box, G-box, and MRE). This suggested that light signals may have significantly impacted the transcriptional control of the PhUGT genes. There were 102, 87, 43, 43, 54, 47, and 55 PhUGT gene promoters that contained, in that order, the ABA-responsive element (ABRE), MeJA-responsive element (CGTCA-motif), salicylic acid-responsive element (TCA-element), low-temperature-responsive element (LTR), drought-inducibility-responsive element (MBS), defense and stress-responsive element (TC-rich repeats), and auxin-responsive element (TGA-element). These findings imply that most of the PhUGT genes might be involved in plant hormones and stress responses. 

### 2.5. PhUGT Gene Expression Study in Five Tissues

The expression pattern of PhUGT genes in the flower, stem, leaf, root, and bud was investigated using RNA-seq. The findings revealed that 37 genes were not found in any tissue, but 67 PhUGT genes were expressed in all five tissues (Figure 4). Ten, 13, 17, 21, and 12 PhUGT genes showed high levels of expression in the flower, stem, leaf, root, and bud, respectively, indicating diverse activities in the various tissues. Additionally, there were no connections between the expression pattern and the phylogenetic groups, indicating that each PhUGT gene had its expression pattern. 

### 2.6. Subcellular Localization Analysis of PhUGT51

From the phylogenetic tree, we can see that PhUGT19, PhUGT51, and PhUGT78 were clustered into one clade with UGT74E1 and UGT74E2. In view of the fact that the function of UGT74E1 has not been revealed in plants, we selected the PhUGT51, which is closely related to UGT74E1, for further research. *PhUGT51* was cloned and fused to the green fluorescent protein (GFP) gene, which was regulated by the CaMV 35S promoter, to ascertain the subcellular location of the protein. A transient expression test in tobacco leaves was carried out using the constructed expression vector. The GFP fluorescence of the control was seen throughout the entire cell, but PhUGT51-GFP’s green fluorescence was only seen in the nucleus and cell membrane (Figure 5).

### 2.7. Expression Analysis of PhUGT51

To further understand the expression characteristics of *PhUGT51*, the expression level of *PhUGT51* was detected. As we can see, the expression of *PhUGT51* significantly increased upon NaCl treatment at 6, 12, and 24 h, reaching 2.5-fold induction at 6 h and the expression level tended to stabilize. These results demonstrated that *PhUGT51* was related to salt response or regulatory processes (Figure 6).

### 2.8. Phenotypes of Transgenic Petunia Plants Overexpressing PhUGT51

To further evaluate whether overexpression of *PhUGT51* can improve tolerance against salt stresses, we constructed the 35S::PhUGT51 vector, and transformed it into Petunia. Three representative lines, Line 15, Line 17, and Line 29, were selected using qRT-PCR analysis (Figure 7). The results showed that the expression level of *PhUGT51* was upregulated 5–8 times in the overexpressed transgenic plants compared with the control. In the absence of NaCl, the germination rate of transgenic plants and control lines was markedly similar, while in the presence of NaCl, the germination rate was significantly inhibited. When NaCl concentration was 100 mM, the germination rate of control decreased from 12% to 4%, while that of Line 15, Line 17, and Line 29 decreased from 13%, 11%, and 16% to 10%, 8%, and 14%, respectively. When the NaCl concentration increased to 150 mM, the control seeds hardly germinated, while the germination rates of Line 15, Line 17, and Line 29 reached 7.5%, 3%, and 10%, respectively. The above experiments showed that the germination rate of transgenic plants was significantly higher than that of control under high salt concentration, indicating that *PhUGT51* could enhance salt tolerance in Petunia.

## 3. Discussion

The UGT family, a superfamily involved in the glycosylation and modification of plant secondary metabolites, plays a crucial role in regulating the degree of hormone action. The UGT gene family has been discovered in other plants due to the development of genome sequencing across many species but nothing was known about Petunia. We found Petunia UGTs based on genomic data analysis, and we examined their phylogeny, gene structure, and expression patterns to understand the UGT family better. Investigating the functions of the Petunia UGT genes will be substantially aided by the information provided here.

In this study, 129 PhUGT genes were found and classified into 16 groups (A–P) based on phylogenetic analysis with 14 highly conserved groups (A–N) and 2 newly identified groups, O and P in Petunia. This is consistent with studies of Selaginella [22], sorghum [23], and cucumber [24]. We discovered that the more significant number of UGT genes in Petunia compared to Arabidopsis (107 members) was primarily attributable to an expansion within the group’s A and O. Groups A and O comprised 27 and 22 members, representing 21 and 17%, respectively, of the UGT genes in Petunia. Among the examined plant species, groups A and O only expanded in Petunia, whereas Arabidopsis contained 14 and 0 members, respectively. These results suggested that these two groups may play significant roles in the glycosylation of minor compounds in Petunia. However, additional research is necessary.

The number and distribution of functional domains on protein sequences can reveal the structural similarities and differences among gene family members, which can be an essential basis for the phylogenetic classification of gene families. According to the analysis of the conserved domain of amino acid sequence of PhUGT genes, a total of 10 amino acid-conserved motifs were found, among which motif 1 was the Petunia UGT PSPGbox domain. In addition to motif 7, other motifs existed in most of the UGT members, indicating that the sequences of UGT family members had the same conserved motifs. The intron mapping of 129 Petunia UGTs revealed that 44% of members lacked introns, which was less than the number (60%) of maize [7] and the number (58%) of Arabidopsis UGT genes [25], while close to the number (40%) for flax [26]. Additionally, 41% of the UGTs contained one intron. 

Understanding the characteristics of cis-acting promoter elements is the premise for analyzing the characteristics of gene transcriptional regulation and expression patterns. In Petunia, the promoters of UGT members contained various numbers of cis-acting elements. ABRE was the most widely distributed element and was present in 79% of PhUGT members, among which the number of ABRE elements in the PhUGT7 promoter reached five (Figure 3), indicating that the abscisic acid induced the expressions of PhUGT genes. In addition, more than 98% of the UGT gene promoters contained cis-acting elements related to growth and development, and Box 4 was the most widely distributed element, with 25% of UGT gene promoters containing three or more Box 4 elements. Furthermore, 99% of the PhUGT gene promoters had one or more cis-acting elements in response to abiotic stresses (Figure 3). This result indicated that the transcriptional regulations of PhUGT genes were jointly completed by the external environment, plant hormones, and growth and development.

To examine the function of Petunia UGT genes, expression analysis based on RNA-Seq was performed in diverse tissues. As shown in Figure 3, PhUGT gene transcripts abundance was associated with different tissues. Approximately 51.9% (67 of 129) of the PhUGT genes were expressed in all tissues, suggesting the vital roles of PhUGT genes in controlling plant growth and developmental processes. We found that *PhUGT61*, the homolog gene of *AtUGT76C2*, had a higher expression level in roots, which was consistent with a previous report that the overexpression of *UGT76C2* enhanced root growth, which significantly contributed to stress adaptation [27].

To investigate the subcellular localization of UGTs, we generated translational fusion constructs of PhUGT51-GFP under the control of the CaMV35S promoter and introduced them into tobacco leaf protoplasts. We found that PhUGT51 localized to the nucleus and cell membrane. This is different from the subcellular localization of other members of UGT, such as BoaUGT74B1, which was localized in chloroplasts [28], and PbUGT72AJ2, which was localized mainly in the cytomembrane and cytoplasm [29]. It indicates the different functions among UGT family genes

The function of *UGT74E1* has not been reported yet. In this study, we found that, under salt stress, the expression level of *PhUGT51* increased significantly but did not change with the increase in treatment time. This is different from the changing trend of other UGT members under salt stress, such as the expression of *UGT79B2* and *UGT79B3*, which upregulated first and then downregulated when treated with NaCl and reached the highest at 6 h [30]. This indicates that different UGT genes play different roles in regulating salt stress. After salt stress treatment, transgenic plants also showed better growth than the control. These results indicated that the overexpression of *PhUGT51* caused an increase in the salt stress tolerance of Petunia, which was similar to the functions of other members of the UGT family, such as *CrUGT87A1* [31], *UGT79B2/B3* [30], and *MdUGT83L3*, both enhanced salt tolerance [32]. However, whether UGT is involved in other stress resistance processes of plants still requires further experiments to explore.

## 4. Conclusions

In this study, the UGT gene family of petunia was analyzed systematically. A total of 129 members of PhUGT were identified, and the phylogenetic tree, intron/exon structures, motif compositions, cis-acting regions, and expression patterns based on the RNA-seq analysis of PhUGT genes were evaluated to obtain a better insight into the roles of the UGT gene family in Petunia. PhUGT51 was proved to be located in the nucleus and cell membrane. *PhUGT51* can be rapidly induced by salt stress, and its overexpression resulted in an excessive increase in salt tolerance, indicating a crucial role in controlling salt stress. To sum up, our study provides a theoretical basis for revealing the functions of PhUGT family genes and provides a genetic resource for genetic engineering to improve the salt resistance of Petunia.

## 5. Materials and Methods

### 5.1. Plant Growth

Petunia×hybrida cv ‘Mitchell Diploid’ was used as the material in this study. The growth condition was described as in Yao et al. [33].

### 5.2. Identification of UGT Genes in Petunia

The sequences of Arabidopsis UGTs were downloaded from The Arabidopsis Information Resource (TAIR9) (www.arabidopsis.org, accessed on 1 December 2020). The UGT family member sequences of Petunia were retrieved from Solanaceae Genome Resource (https://solgenomics.net/, accessed on 1 December 2020). Delete too long or too short sequences and then use Pfam (http://pfam.xfam.org/, accessed on 1 December 2020) to search the UGT family domain. Remove the sequences that do not contain the PSPG box domain, and finally obtain the Petunia UGT family member sequences. The prediction of the physical and chemical features of PhUGT proteins was conducted according to the method of Yao et al. [33].

### 5.3. Phylogenetic, Motif Distribution and Gene Structure Analysis

A total of 129 PhUGTs and 21 AtUGTs were used to generate a phylogenetic tree. The construction of the phylogenetic tree and analysis of conserved motif distribution and gene structure was conducted according to the method of Yao et al. [33].

### 5.4. Expression Profile Analysis of Petunia UGT Genes

Petunia×hybrida cv ‘Mitchell Diploid’ seedlings that were 70 days old were used as experimental materials. Roots, stems, leaves, buds, and flowers were harvested for RNA extraction. The RNA was extracted and sent to Beijing Novogene Technology Co. Ltd. Company (Beijing, China) for transcriptome sequencing. Subsequently, Fragments Per kb per Million reads (FPKM) values were calculated to evaluate the gene expression values, and the heatmap was generated using the ClustVis tool (https://biit.cs.ut.ee/clustvis/, accessed on 6 December 2020).

The forty-day-old Petunia plants were chosen and irrigated with salt (300 mM NaCl). Petunia leaves were harvested at 0, 6, 12, and 24 h after treatment for further gene expression analysis. The expression level of *PhUGT51* was detected by qRT-PCR with primers UGT51-RT-F/UGT51-RT-R, and *PhGAPDH* was employed as the internal control (Appendix A). The comparative CT method was used to calculate the relative expression level [34]. Three technical replications and 3 biological replications for each sample were used to analyze the expression of *PhUGT51*.

### 5.5. Analysis of the Cis-Regulatory Elements in PhUGT Genes

The 1700 bp upstream flanking sequences from the transcription start site of each putative PhUGT gene were downloaded. Then the conserved cis-elements in the promoter regions of PhUGT genes were investigated using PlantCARE (http://bioinformatics.psb.ugent.be/webtools/plantcare/html/, accessed on 8 December 2020).

### 5.6. Subcellular Localization

The coding regions of *PhUGT51* were amplified by polymerase chain reaction (PCR) with primers UGT51-F/UGT51-R and then fused to the pSuper1300-eGFP plant expression vector by homogenous recombination method (Appendix A). Subcellular localization was carried out according to the method of Yao et al. [33].

### 5.7. Overexpression of PhUGT51 and Phenotype Analysis in Petunia

35S::PhUGT51 was transformed to Petunia×hybrida cv ‘Mitchell Diploid’ according to the method of Guo et al. [35]. The expression level of *PhUGT51* in transgenic plants was detected by qRT-PCR. The seeds of Petunia were sterilized in 75% ethanol for 30 s, followed by 2.5% NaClO for 6 min, and finally washed with sterile distilled water 3 times. After stratification at 4 °C in darkness for 4 d, the seeds were put on plates containing 1/2 Murashige and Skoog (MS) medium supplemented with NaCl (0, 100 mM, 150 mM, 200 mM). After 20 days, photos were taken to observe the phenotype.

## Figures and Tables

**Figure 1 plants-11-02434-f001:**
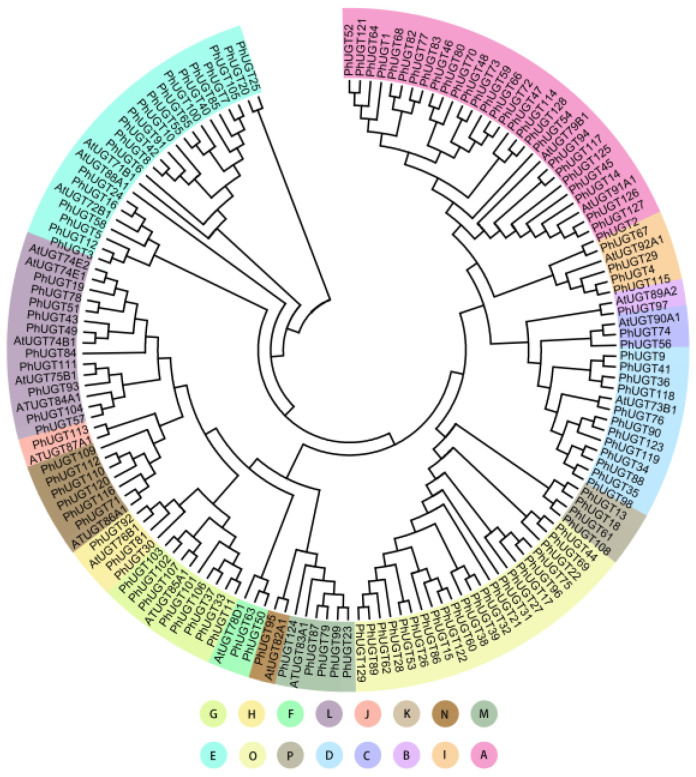
Phylogenetic tree of the UGT family.

**Figure 2 plants-11-02434-f002:**
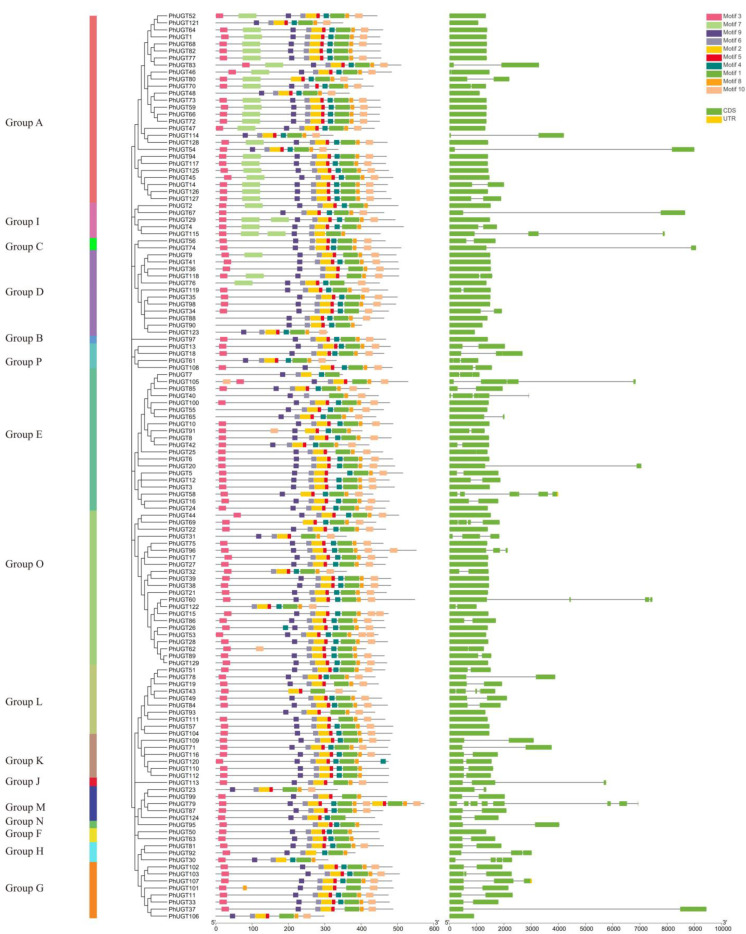
Conserved protein motifs and gene structure of PhUGT family members.

**Figure 3 plants-11-02434-f003:**
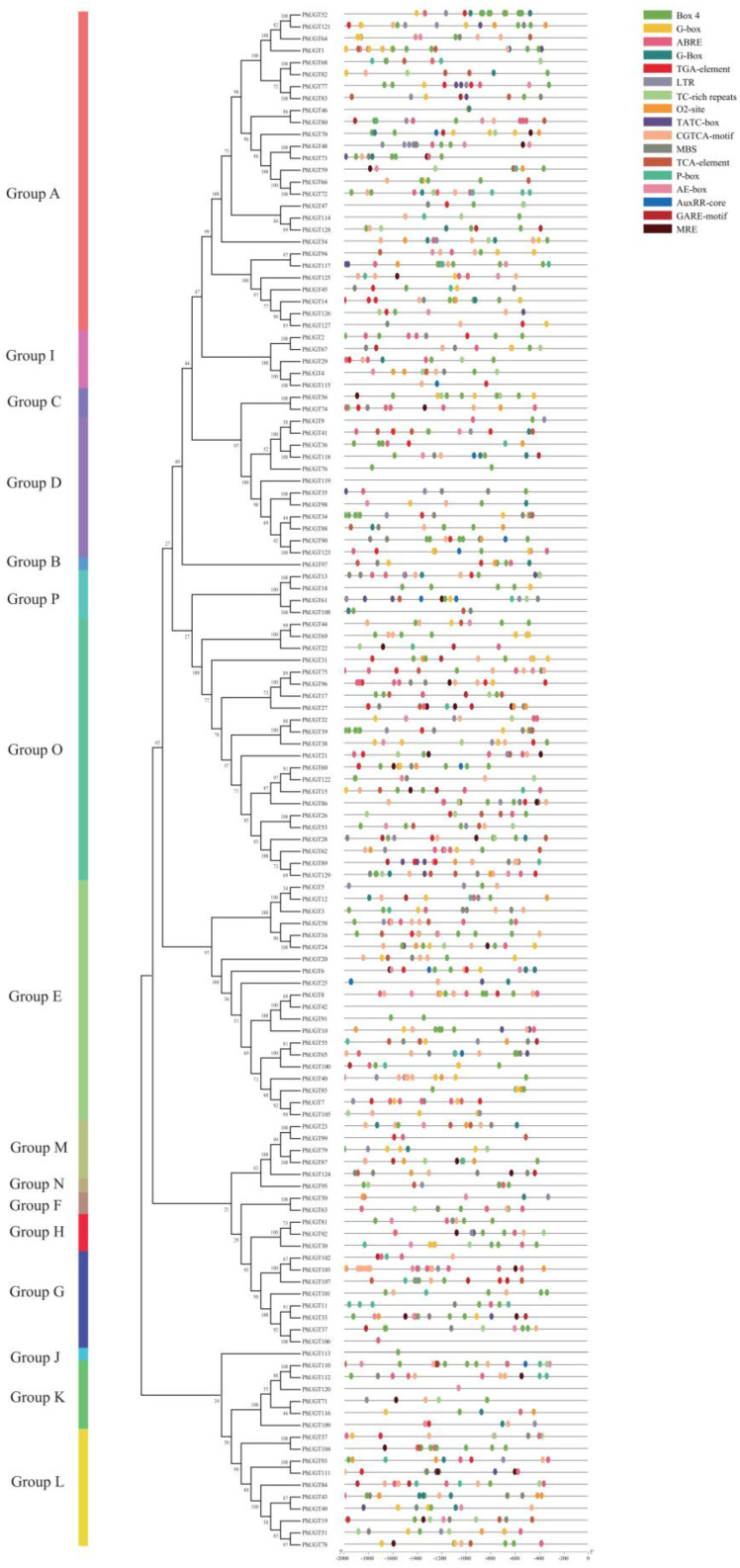
Predicted cis-acting elements in the promoter regions of PhUGT genes.

**Figure 4 plants-11-02434-f004:**
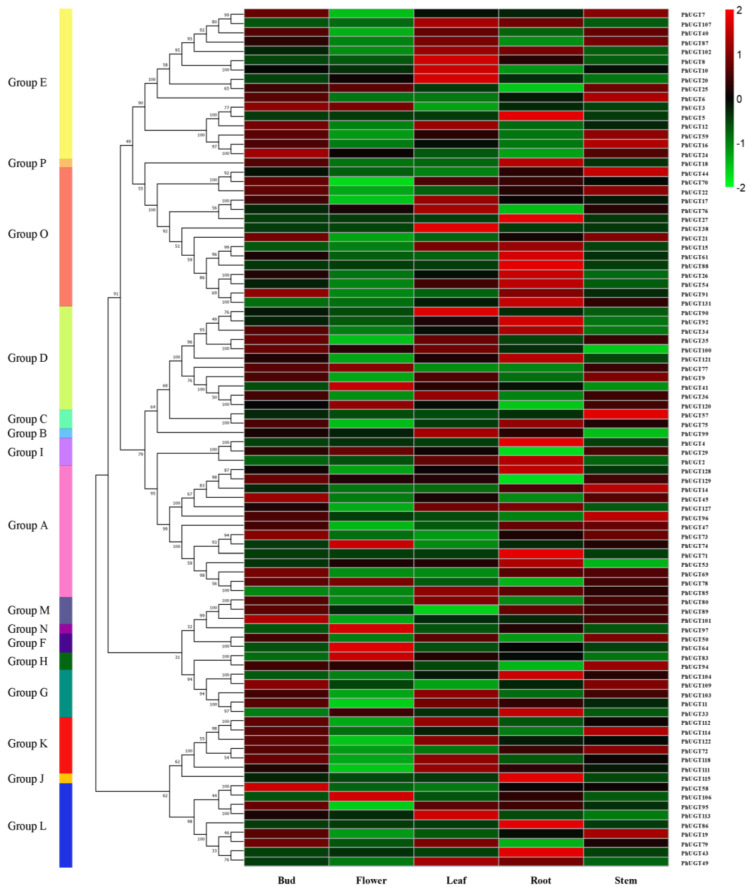
Expression heat map of different tissues of Petunia UGT family members.

**Figure 5 plants-11-02434-f005:**
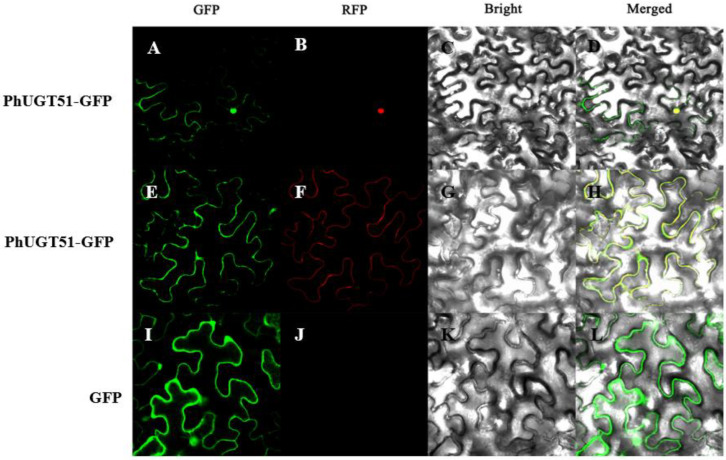
Subcellular localization analysis of PhUGT51. (**A**,**E**,**I**) Green fluorescence pictures of PhUGT51-GFP protein and GFP (control). (**B**,**J**) Red fluorescence pictures of a nucleus marker. (**F**) Red fluorescence picture of a cell membrane marker. (**C**,**G**,**K**) Bright field pictures of PhUGT51-GFP protein and control. (**D**,**H**,**L**) The combined pictures of PhUGT51-GFP protein and control.

**Figure 6 plants-11-02434-f006:**
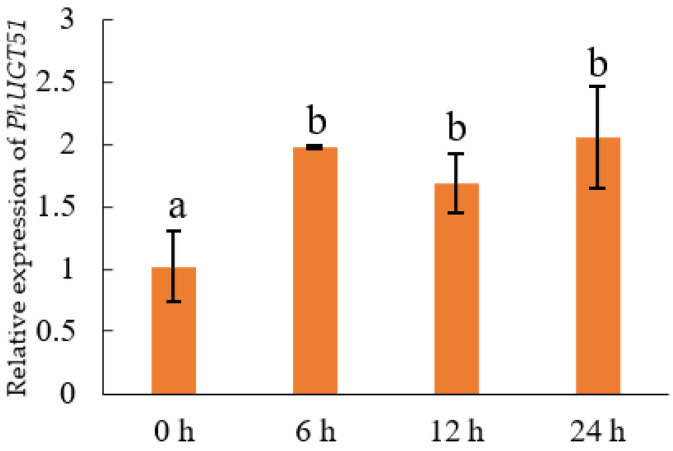
Quantitative real-time PCR (qRT-PCR) analysis of *PhUGT**51* after various NaCl treatment periods. Lowercase letters indicate significant differences.

**Figure 7 plants-11-02434-f007:**
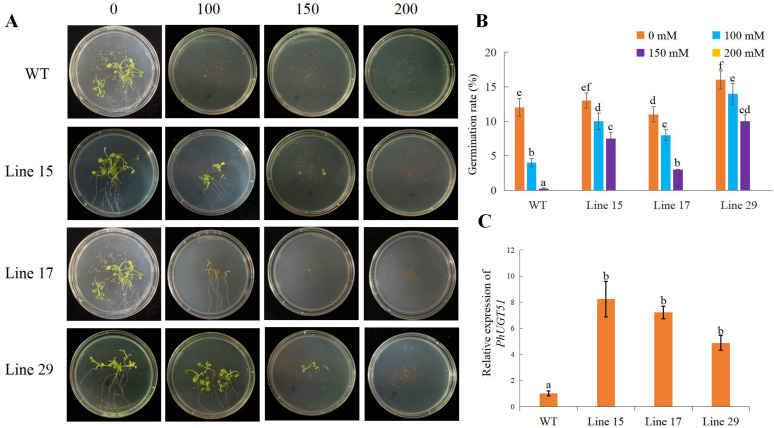
Analysis of the phenotypes of Petunia plants overexpressing *PhUGT51*. (**A**) Comparisons between the phenotypes of control and transgenic plants that overexpress *PhUGT51*. Different lines overexpressing *PhUGT51* are Line 15, Line 17, and Line 29. (**B**) The germination rate is displayed. (**C**) qRT-PCR was used to detect the expression level of *PhUGT51*. Fifteen plantlets from each of the three samples were averaged (SE). Lowercase letters indicate significant differences.

## Data Availability

All the datasets included in this study have been presented within the manuscript and/or as Appendix A.

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
