# Peer review of "Genome-Wide Analysis of UGT Genes in Petunia and Identification of *PhUGT51* Involved in the Regulation of Salt Resistance"

_plants, 2022, doi:10.3390/plants11182434_

Round 1

Reviewer 1 Report

Dear authors,

The manuscript is nicely written and the experiments nicely designed and conducted. Meanwhile to suggest your manuscript for acceptance I recommend that you address the following points:

1. Introduction section: Please discuss more widely function and relationship between structure and function of UDP-glycosyltransferase in plants.

2. Results section: please improve the resolution of the figures.

I will be happy to review an improved version of the manuscript that I can recommend for publication

Best regards

Author Response

  1. We have revised the  Introduction section.
  2. We have improved the resolution of the figures.

Reviewer 2 Report

1. In Figure 1, 2, 3 and 4, there is a lack of description of relevant results, especially for PhUGT74E1. Family analysis is not a simple stack of data.

2. Please explain why PhUGT74E1 was chosen as the research object.

3. Please attach a detailed list of the PhUGT gene family members used in FASTA format.

4. In Figure 5F, it is not like the chloroplast marker, because chloroplasts usually appear as breakpoints, please do the plasma-wall separation experiment to exclude the possibility of cell membrane.

5. All data are missing ANOVA, please rewrite and provide original data and data processing.

6. Please provide the results of positive tests for transgenic plants, at least the results of PCR.

7. In Figure 7, the germination rate experiment was not standard. Germination experiments require a neat arrangement of seeds. Uneven seed accumulation can also affect germination. Please refer to the relevant literature of Plant Cell to standardize the experiment.

Author Response

  1. In Figure 1, 2, 3 and 4, there is a lack of description of relevant results, especially for PhUGT74E1. Family analysis is not a simple stack of data.

We have revised these places.

  1. Please explain why PhUGT74E1 was chosen as the research object.

  We have added these descriptions in 2.6

  1. Please attach a detailed list of the PhUGT gene family members used in FASTA format.

We have added the tables. we forgot to upload them when submitting.

  1. In Figure 5F, it is not like the chloroplast marker, because chloroplasts usually appear as breakpoints, please do the plasma-wall separation experiment to exclude the possibility of cell membrane.

PhUGT74E1 (PhUGT51) was localized on the cell membrane and nucleus, and the red was the marker of the cell membrane. There was a mistake in the paper.

  1. All data are missing ANOVA, please rewrite and provide original data and data processing.

We have added them.

  1. Please provide the results of positive tests for transgenic plants, at least the results of PCR.

We performed qRT-PCR of PhUGT74E1 (PhUGT51) in transgenic and control plants, please see Fig 7C

  1. In Figure 7, the germination rate experiment was not standard. Germination experiments require a neat arrangement of seeds. Uneven seed accumulation can also affect germination. Please refer to the relevant literature of Plant Cell to standardize the experiment.

We have repeated this experiment three times. Sometimes the seeds are very sparse, and the trend of germination rate is basically the same. It's just that the photos are not very beautiful, so we chose the flat plate with dense seeds to take photos. If we really need to do it again, we will repeat it according to the reviewers' suggestions.

Reviewer 3 Report

This research identified UGT proteins from the Petunia genome database and orderly named. Besides, a comprehensive analysis of PhUGT genes was accomplished; including their phylogenetic relationship, exon-intron structure, conserved domains, cis-regulatory analysis and expression patterns in various tissues, subcellular localization and function of PhUGT74E1 were investigated. The manuscript is well structured and well discussed. However, some points should be checked and corrected before its acceptance in this journal. 

Therefore, according to my comments, I recommended the publication of the paper after minor revision.

[1]   The study's background should be clearly stated. Describe the introduction and review of the work (Please add more information).

[2]   Conclusions should be improved. The authors should add the significance of this research and its potential practical application.

[3]   The MS English needs to be improved. The article's English must be carefully checked for grammatical errors.

Author Response

  1. The study's background should be clearly stated. Describe the introduction and review of the work (Please add more information).

We have revised the introduction part.

  1. Conclusions should be improved. The authors should add the significance of this research and its potential practical application.

We have revised the conclusion part.

  1. The MS English needs to be improved. The article's English must be carefully checked for grammatical errors.

We have carefully revised the full text.
